# FOURIER SENSITIVITY AND REGULARIZATION OF COMPUTER VISION

## ABSTRACT

Deep neural networks for computer vision are highly accurate but their out-of-distribution generalization performance remains a major challenge, so understanding their failure modes is an important step towards improving their robustness. Intriguingly, recent work has shown that deep neural networks latch on to the Fourier statistics of training data, show increased sensitivity to certain Fourier-basis directions in the input and are not robust when Fourier-statistics shift unfavorably at test time. Understanding and modifying this Fourier-sensitivity of computer vision models may help improve their robustness, hence, in this paper we study the frequency sensitivity characteristics of deep neural networks using a principled approach. We first propose a general measure of a computer vision model's *spatial frequency sensitivity* based on its input-Jacobian represented in the Fourier-basis. When applied to deep neural networks, we find that standard minibatch training consistently leads to increased sensitivity towards particular spatial frequencies independent of network architecture. We further propose a family of *spatial frequency regularizers* based on our proposed measure to induce specific spatial frequency sensitivities in a model. In various generalization tests involving shifts in the Fourier-statistics of the data, we find that deep neural networks trained with our proposed regularizers obtain significantly improved classification accuracy while maintaining high accuracy on in-distribution test images.

## 1 INTRODUCTION

While deep neural networks (DNN) achieve remarkable performance on many challenging image classification tasks, they can suffer significant drops in performance when evaluated on out-of-distribution (o.o.d.) data. Intriguingly, this lack of robustness has been partially attributed to the frequency characteristics of data shifts at test time in relation to the frequency sensitivity characteristics of the model (Yin et al., 2019; Jo & Bengio, 2017). Distinct spatial frequencies in images contain features at different spatial scales; low spatial frequencies (LSF) carry global structure and shape information in a scene whereas high spatial frequencies (HSF) carry local information such as edges and borders of objects in a scene (Kauffmann et al., 2014); in fact, spatial frequencies are differentially processed in distinct channels of the visual cortex in the brain to learn features at different scales (Appendix B). When information is destroyed or corrupted in frequency bands that a model relies on, performance suffers. Hence, understanding the spatial frequency sensitivity of a DNN can help us characterise the features it relies on to make predictions.

DNNs have been demonstrated to be sensitive to Fourier-basis directions in the input (Tsuzuku & Sato, 2019; Yin et al., 2019) both empirically and using theoretical analysis of linear convolutional networks (Tsuzuku & Sato, 2019). In fact, the existence of so-called "universal adversarial perturbations" (Moosavi-Dezfooli et al., 2017), simple semantics-preserving distortions that can degrade models' accuracy across inputs and architectures, is attributed to this structural sensitivity and the use of convolution operations. Yin et al. (2019) also showed that many natural and digital image corruptions that degrade model performance may also be targeting this vulnerability. The Fourier-characteristics of adversarial examples are also known to closely match the Fourier-sensitivity of models (Yin et al., 2019). Hence, understanding and modifying Fourier-sensitivity can aid efforts to improve model robustness. While this Fourier-sensitivity has been studied empirically, the precise definition and measurement of a computer vision model's *spatial frequency sensitivity* still lacks a

rigorous approach across studies. In addition, there has been no principled method to specifically modify the spatial frequency sensitivity of a model. Existing works have heuristically applied filters on convolution layer parameters (Wang et al., 2020; Saikia et al., 2021) and data augmentations (Yin et al., 2019) to modify a model's frequency sensitivity.

In this work, we propose a novel and rigorous measure of a deep neural network's *spatial frequency sensitivity* using the input-Jacobian represented in the Fourier-basis and show that deep neural networks demonstrate consistent spatial frequency sensitivities across samples, an observation that suggests DNNs are more likely to consistently use some frequencies more than others, and has implications for robustness. In addition, using our proposed measure, which is differentiable with respect to model parameters, we propose a novel family of *spatial frequency regularizers* to directly induce specific frequency sensitivities in a model. We hypothesize and show in empirical evaluations that spatial frequency regularization can modify the frequency sensitivity characteristics of computer vision models and can significantly improve the generalization performance of models on o.o.d. datasets where the Fourier-statistics are unfavorably shifted.

In summary, the main contributions of this work are as follows:

1. We propose a novel and rigorous measure of a model's **spatial frequency sensitivity** based on its input-Jacobian represented in the Fourier-basis

2. We propose a novel family of **spatial frequency regularizers** to directly induce specific spatial frequency sensitivities

3. We demonstrate that spatial frequency regularization can significantly improve generalization performance on **out-of-distribution** data where Fourier-statistics are unfavorably shifted

## 2 RELATED WORK

**Characterising frequency sensitivity:** Yin et al. (2019); Tsuzuku & Sato (2019) characterised the Fourier characteristics of trained CNNs using perturbation analysis of their test error under Fourier-basis noise. They showed that a naturally trained model is most sensitive to all but low frequencies whereas models adversarially trained (Madry et al., 2018) models are sensitive to low-frequency noise. They further showed that these Fourier characteristics relate to model robustness on corruptions and noise, with models biased towards low frequencies performing better under high frequency noise and vice versa. Abello et al. (2021) took a different approach by measuring the impact on accuracy of removing individual frequency components from the input using filters whereas Ortiz-Jimenez et al. (2020) computed the margin in input space along basis directions of the discrete cosine transform (DCT). Wang et al. (2020) made observations about the Fourier characteristics of CNNs in different training regimes including standard and adversarial training by evaluating accuracy on band-pass filtered data. In contrast to these disparate approaches, in this work, we propose a rigorous measure of a model's *spatial frequency sensitivity*.

**Regularizing frequency sensitivity:** Yin et al. (2019) proposed adversarial training (Madry et al., 2018) and gaussian noise augmentations as methods that induce a low-frequency sensitivity. Wang et al. (2020) proposed smoothing convolution filter parameters to induce a low-frequency sensitivity in models. We note that, in general, techniques that apply filters on convolutional parameters to affect a model's frequency sensitivity do not take into account complex operations such as non-linearities, pooling and other transformations that often follow convolutional layers that can modify as well as undo the effects of such filters. In addition, data augmentations do not provide precise control over the Fourier-sensitivity of a model. In this work, we propose a family of *spatial frequency regularizers* that can precisely modify the overall spatial frequency sensitivity of any differentiable model.

**Jacobian regularization:** Methods that regularize the Jacobian of a model's output-logits or loss with respect to its input can broadly be divided into two types; one regularizes the norm of the input-Jacobian and the other regularizes its direction or directional derivatives. Drucker & Le Cun (1991) proposed a method that penalized the norm of the input-Jacobian to improve generalization, more recently this has been explored to improve robustness to adversarial perturbations (Ross & Doshi-Velez, 2018; Jakubovitz & Giryes, 2018; Hoffman et al., 2019). Simard et al. (1992) proposed

"Tangent Prop", which minimized directional derivatives of classifiers in the direction of local input-transformations (e.g. rotations, translations; called "tangent vectors") to reduce model sensitivity to such transformations. Czarnecki et al. (2017) proposed Sobolev training of neural networks to improve model distillation by matching the input-Jacobian of the original model. Regularizing the direction of the input-Jacobian has also been used to improve adversarial robustness (Chan et al., 2020). In the present work, we regularize Fourier-components in the input-Jacobian to modify the spatial frequency sensitivity of models. As such, we are directly modifying the direction of the input-Jacobian instead of its norm.

## 3 METHODS

**Preliminaries:** We introduce all relevant definitions and notations before describing the proposed methods. Consider an image classification task with input images $x$, target labels $y$, and the standard cross-entropy loss function $\mathcal{L}_{\mathrm{CE}}$. Let $f$ denote any differentiable model, $\mathcal{F}(\cdot)$ the unitary 2D discrete Fourier transform (DFT), $\mathcal{F}^{-1}(\cdot)$ its inverse, and $\mathcal{F}^{-1^*}(\cdot)$ the adjoint of the inverse-Fourier transform, and let $x_f$ denote the Fourier space representation of the input, i.e. $x_f = \mathcal{F}(x)$. We denote the input-Jacobian in the standard input basis as $\frac{\partial \mathcal{L}_{\mathrm{CE}}}{\partial x}$, and $\frac{\partial \mathcal{L}_{\mathrm{CE}}}{\partial x_f}$ as the input-Jacobian in the Fourier-basis. Let N denote the height of the input images $x$ (although not necessary, all images used in this work are square). The zero-shifted 2D-DFT of the input-Jacobian is denoted $F = \mathcal{F}(\frac{\partial \mathcal{L}_{\mathrm{CE}}}{\partial x})$. Since the input-Jacobian typically has three color channels, they are averaged before computing the 2D-DFT. Fourier coefficients in $F$ are complex numbers with real and imaginary components; $F(u, v) = Real(u, v) + Imag(u, v)$, where $(u, v)$ are indices of coefficients. The *power* in a coefficient is its squared amplitude, $P(u, v) = |F(u, v)|^2 = Real(u, v)^2 + Imag(u, v)^2$ and the matrix of powers is denoted $P$ (power-matrix). Each coefficient has a radial distance $r(u, v)$ from the centre of the matrix, $r(u, v) = d((u, v), (c_u, c_v))$, where $(c_u, c_v)$ denotes the centroid of $P$ and $d(\cdot, \cdot)$ is Euclidean distance. Distinct radial distances, rounded to the nearest integer, of coefficients in the matrix are the set of integers $\{1, \ldots, N/\sqrt{2}\}$ and correspond to low to high spatial frequency bands, the highest spatial frequency being limited by the Nyquist frequency. We denote $P_{Total}$ as the total power in $P$, excluding the zero-frequency coefficient, $P_{Total} = \sum_{r(u,v)>=1} P(u, v)$. Similarly, we define $\tilde{P}_{Total}$ as the total power in $P$ excluding the zero-frequency coefficient as well as coefficients with radial distance $r(u, v) > N/2$, i.e. outside the largest circle inscribed in the power-matrix $P$; $\tilde{P}_{Total} = \sum_{1<=r(u,v)<=N/2} P(u, v)$ (please see Figure 5 in Appendix A.1 for an illustration). We denote $P_k$ as the total power at radial distance $k$ normalized by $P_{Total}$, $P_k = \frac{1}{P_{Total}} \sum_{r(u,v)=k} P(u, v)$ and $\tilde{P}_k$ as total power at radial distance $k$ normalized by $\tilde{P}_{Total}$ instead, $\tilde{P}_k = \frac{1}{\tilde{P}_{Total}} \sum_{r(u,v)=k} P(u, v)$.

### 3.1 SPATIAL FREQUENCY SENSITIVITY OF A MODEL

In this section, we define the proposed ***spatial frequency sensitivity (SFS)*** of any differentiable model using its input-Jacobian represented in the Fourier-basis. As the input-Jacobian provides the direction of highest input sensitivity, we show below that its Fourier transform, which is simply a change of basis, provides the model's sensitivities to spatial frequency components in the input. The input-Jacobian in the Fourier-basis, $\frac{\partial \mathcal{L}_{\mathrm{CE}}}{\partial x_f}$, comprises the sensitivities of the model with respect to individual frequencies in the input and we obtain this by simply computing the Fourier transform of the input-Jacobian $\frac{\partial \mathcal{L}_{\mathrm{CE}}}{\partial x}$. In order to justify this, consider the computation graph where the input $x$ is mapped to a scalar loss via a model $f$ and loss function $\mathcal{L}_{\mathrm{CE}}$ (Figure 2). We introduce an implicit operation (shown in red) that maps the Fourier space representation of the input, $x_f$, to the standard input $x$ through the inverse Fourier-transform; $x_f \xrightarrow{\mathcal{F}^{-1}} x$, so in order to compute the input-Jacobian in the Fourier-basis, $\frac{\partial \mathcal{L}_{\mathrm{CE}}}{\partial x_f}$, we must differentiate through this *implicit* operation in the forward graph. Since the inverse-Fourier transform is a unitary operator, its adjoint is also its inverse, i.e. $\mathcal{F}^{-1^*} = (\mathcal{F}^{-1})^{-1} = \mathcal{F}$. Hence, following the chain rule for complex operators, $\frac{\partial \mathcal{L}_{\mathrm{CE}}}{\partial x_f}$ is simply the Fourier transform of the input-Jacobian $\frac{\partial \mathcal{L}_{\mathrm{CE}}}{\partial x}$.

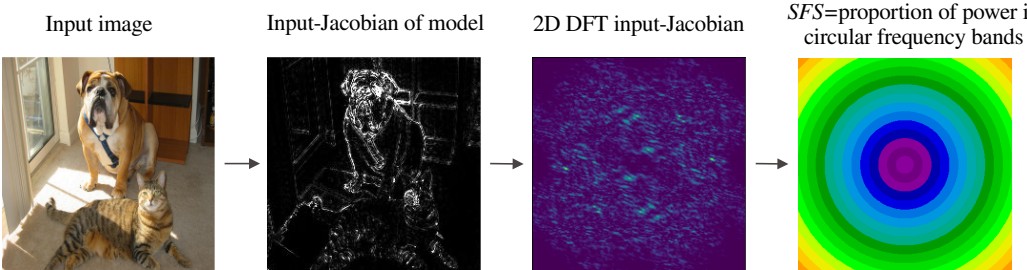

| Input image | Input-Jacobian of model | 2D DFT input-Jacobian | *SFS*=proportion of power in circular frequency bands |

Figure 1: Illustration of steps to compute the *spatial frequency sensitivity* with respect to a single input image. The input-Jacobian of the model is Fourier-transformed to obtain sensitivities with respect to frequency components followed by computing the proportion of power in low to high (small to large radius) frequency bands.

Figure 2: Computation graph to obtain the input-Jacobian in Fourier-space. The symbols in red represent an implicit map of the input from Fourier-space to the standard input space. The dashed arrows denote the backward graph operations to compute the input-Jacobian in Fourier space.

$$\frac{\partial \mathcal{L}_{\text{CE}}}{\partial x_f} = \mathcal{F}^{-1*}\left(\frac{\partial \mathcal{L}_{\text{CE}}}{\partial x}\right) = \mathcal{F}\left(\frac{\partial \mathcal{L}_{\text{CE}}}{\partial x}\right)$$

Hence, even though we do not explicitly use the Fourier representation of the input, this shows that the Fourier transform of the input-Jacobian provides us the sensitivity of the model with respect to spatial frequencies of the input. In fact, similar results hold for any unitary operation on the input-Jacobian, such as the discrete cosine transform (DCT). Interested readers can replace the DFT ($\mathcal{F}$) with other such transforms if they would like to understand the sensitivity of a model with respect to other components in the input. We now define the spatial frequency sensitivity $f_{SFS}(x,y)$ with respect to an individual input $(x,y)$ as,

$$f_{SFS}(x,y) = \{P_1, \ldots, P_{N/\sqrt{2}}\}$$

where $P_k$ is the proportion of total power in Fourier coefficients at radial distance $k$ in the power matrix $P$ of the Fourier-transformed input-Jacobian. The overall spatial frequency sensitivity of a model is defined as the expectation of $f_{SFS}(x,y)$ over the data distribution, i.e. $f_{SFS}(\cdot; \theta) = \mathbb{E}_{(x,y)\sim p}[f_{SFS}(x,y)]$ (please see Figure 1 for an illustration and Algorithm 1 in Appendix A.1). We note that although we use the Jacobian of the loss function to estimate the *SFS* of a model, this formulation is valid for other output functions and tasks as well. An alternative approach that does not require image labels is to compute the Jacobian of the model's output logits or softmax probabilities. In addition, while the spatial frequency sensitivity assumes the input is spatial 2D data (e.g. images), we can extend this approach to any n-dimensional data by using the n-dimensional Fourier-transform as well as to other computer vision tasks.

## 3.2 SPATIAL FREQUENCY REGULARIZATION

In this section, we propose a novel family of **spatial frequency regularizers** that can modify the *spatial frequency sensitivity*, as defined in Section 3.1, of a model directly. As all computations involved in the *SFS* of a model (Algorithm 1 in Appendix A.1) are differentiable with respect to its parameters, we compute a loss function, $\mathcal{L}(x,y) = \mathcal{L}_{\text{CE}}(x,y) + \lambda_{\text{SFS}}\mathcal{L}_{\text{SFS}}(x,y)$, where $\mathcal{L}_{\text{SFS}}$ is the proposed loss that can be used to induce a specific *SFS* and $\lambda_{\text{SFS}}$ is a hyperparameter. As $\mathcal{L}_{\text{SFS}}$ is defined on the *SFS*, which is a function of the input-Jacobian, optimizing it requires an additional backpropagation step to compute its derivatives with respect to model parameters, similar

to other Jacobian-regularization methods. We now define $\mathcal{L}_{\text{SFS}}$ for three different SFS regularizers; $SFS \in \{LSF, MSF, ASF\}$. LSF regularization trains a model to be insensitive to medium and high spatial frequencies and MSF regularization trains a model to be insensitive to low and high spatial frequencies. We achieve this by penalizing the proportion of total power, $P_k$, in frequency bands. ASF regularization trains a model to be equally sensitive to all spatial frequency (ASF) bands. The motivation behind ASF regularization model is to encourage a model to be sensitive to multiple frequency bands instead of being concentrated in a single frequency range. Hence, the ASF-regularizer loss is the negative entropy of the distribution of power over spatial frequency bands.

$$
\mathcal{L}_{\text{SFS}}(x, y) := \begin{cases} \sum\limits_{k > N/6} P_k, & \text{if SFS=LSF} \\ \sum\limits_{k < N/6, k > N/3} P_k, & \text{if SFS=MSF} \\ \sum\limits_{k=1}^{k=N/2} \tilde{P}_k \log \tilde{P}_k, & \text{if SFS=ASF} \end{cases}
$$

The definitions of frequency ranges are based on equally dividing the largest circle inscribed in the power-matrix $P$ into equal parts (Figure 5 in Appendix A.1). For ASF-regularization, very high frequency bands ($r(u,v) > N/2$) are excluded. Interested readers may modify these loss functions as they see fit for their particular application.

## 4 EXPERIMENTS

In this section we explore the *SFS* of models as well as explore the effects of spatial frequency regularization by evaluating on datasets where the Fourier-statistics of predictive features are naturally or artificially altered (Sections 4.2, 4.4). We also make some observations about the differences between spatial frequency regularization and training on Fourier-filtered training data (Section D.3). We benchmark against methods that have been proposed to modify the frequency sensitivity of models such as adversarial training and Gaussian Noise augmentation (Yin et al., 2019) to induce low-frequency sensitivity. AugMix (Hendrycks et al., 2020) is also benchmarked as it was shown to achieve state-of-the-art performance on common corruptions. We also trained models on band-pass filtered datasets to demonstrate that SFS regularization is not equivalent to training on Fourier-filtered data (discussion relegated to Appendix D.3).

**Experimental Setup:** We first describe the experimental settings for all experiments that follow. We trained the ResNet50 architecture, unless stated otherwise. For CIFAR10 and CIFAR100 (Krizhevsky & Hinton, 2009), we trained all models for 150 epochs using stochastic gradient descent (SGD) with momentum (0.9), an initial learning rate of 0.1 decayed by a factor of 10 every 50 epochs, weight decay parameter equal to 5e-4 and batch size equal to 128. For SVHN (Netzer et al., 2011), we trained models for 40 epochs using Nesterov momentum with an initial learning rate of 0.01 and momentum parameter 0.9. The training batch size was 128, the L2 regularization parameter was 0.0005 and we decayed the learning rate at epochs 15 and 30 by dividing by 10. Standard data augmentations random-crop, random-horizontal-flip, random-rotation, and color-jitter were used during training. We trained spatial frequency regularized models using Algorithm 2 2 in Appendix A.1, and $\lambda_{\text{SFS}} = 1$ for all models. We observed a simple trade-off between this hyperparameter and clean accuracy (higher values decreased clean accuracy more). Values less than 1 did not always achieve the desired *SFS* in the evaluated datasets. For adversarial training (AT), we used PGD $\ell_2$ attacks ($\epsilon = 1$, attack-steps $= 7$ attack-lr $= 1/7$). To train models with Gaussian noise data augmentation, we added i.i.d. Gaussian noise $\mathcal{N}(0, \sigma^2)$ to each pixel in all the images in the training set ($\sigma = 0.1$). We used the *robustness* (Engstrom et al., 2019) library to train models as well as retrieve pre-trained ImageNet models of various architectures. We trained AugMix models using the original code shared by the authors (`https://github.com/google-research/augmix`). Please see Section D.2 in Appendix D for details about band-pass filters.

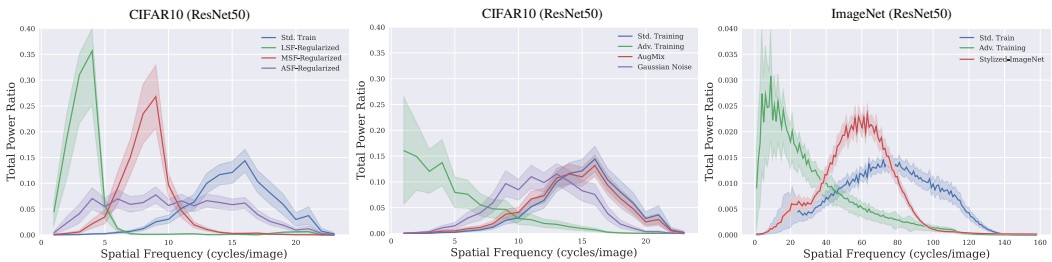

Figure 3: SFS of ResNet50 models trained by methods on CIFAR10 and ImageNet. Computed on 1k randomly selected samples from respective validation sets. The shaded region represents two standard deviations.

### 4.1 CNNs DISPLAY CONSISTENT SPATIAL FREQUENCY SENSITIVITY ACROSS SAMPLES

We now visualize the *spatial frequency sensitivity (SFS)*, as defined in Section 3.1, of models trained on CIFAR10 and ImageNet (Figure 3). We note that training biases models to be sensitive to some frequencies more than others and this bias is consistent across samples (the error bars in Figure 3 represent deviation across samples). This consistency also suggests that, once trained, CNNs are more likely to use some frequencies more than others and may explain their tendency to suffer performance drops when evaluated on data with different Fourier-statistics. CIFAR100 models have very similar *SFS* to those of CIFAR10 (plots relegated to Appendix C.2) and are also most sensitive to mid-to-high spatial frequencies. In contrast, standard training on SVHN leads to a low-frequency sensitivity, which suggests a strong dataset dependence of spatial frequency sensitivity. Please refer to Appendix C.3 for SFS plots of models trained on SVHN.

ImageNet models are sensitive to a wide range of the frequency spectrum with peak sensitivity to mid-range frequencies. We also observed consistency in the spatial frequency sensitivity of various convolutional architectures trained on ImageNet (Appendix C.1). Adversarially trained models (Madry et al., 2018) are most sensitive to low spatial frequencies across datasets and architectures which suggests they rely on coarse global features, in agreement with observations made in prior work. Training on Stylized-ImageNet, proposed by Geirhos et al. (2019) to train shape-biased in models, induces a lower peak-spatial-frequency-sensitivity and as well as lower sensitivity to high frequencies compared to training on ImageNet (Appendix C.2), which reflects the increased shape-bias of such models. Similarly, a model trained on low-pass Fourier-filtered training data correspondingly has high sensitivity to low-frequencies in comparison to the baseline model (Appendix C.2). Gaussian noise augmentation $\mathcal{N}(0, \sigma^2; \sigma = 0.1)$, decreases the high-frequency sensitivity of the model (Figure 3) compared to baseline. The *SFS* of the AugMix (Hendrycks et al., 2020) trained model was not significantly different from baseline, which suggests it finds more robust features in similar frequency bands using data augmentations.

Interestingly, we also note that models trained on corrupted versions of the CIFAR10 training set (e.g. blurred or JPEG-compressed) can display different spatial frequency sensitivities to a model trained on clean CIFAR10 images (please see Appendix C.4 for plots). For example, models trained on images with severe noise corruptions (gaussian, shot and speckle) display increased sensitivity to lower spatial frequencies. Models trained on highly gaussian-blurred, glass-blurred, JPEG-compressed and pixelated images also display higher sensitivity to lower spatial frequencies. These changes reflect the shift in the Fourier-statistics of predictive features in corrupted images.

### 4.2 PERFORMANCE UNDER FOURIER FILTERING

Jo & Bengio (2017) showed that DNNs have a tendency to rely on superficial Fourier-statistics of their training data. In the vein of generalization evaluations they performed, we generate Fourier-filtered CIFAR10/CIFAR100 validation images using radial masking in frequency space (please see Appendix D.1 for examples). A mask radius $r$ determines Fourier components that are preserved with larger radii preserving more components. We use $(c_u, c_v)$ to denote the centre of the mask and

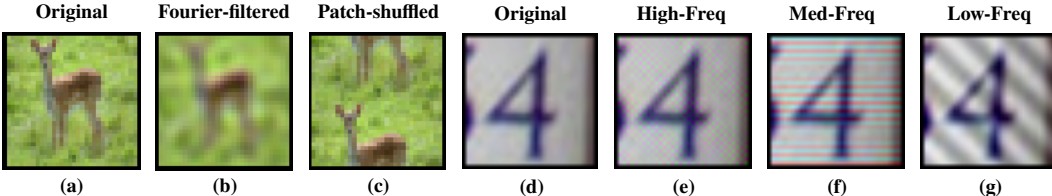

|  Original | Fourier-filtered | Patch-shuffled | Original | High-Freq | Med-Freq | Low-Freq |
|:--:|:--:|:--:|:--:|:--:|:--:|:--:|
| (a) | (b) | (c) | (d) | (e) | (f) | (g) |

Figure 4: Examples (b): CIFAR10 Fourier-filtered (Section 4.2) (c): CIFAR10 Patch-shuffled (Section 4.5). (e) - (g): Fourier-noise distortions on SVHN (Section 4.3). More in Appendix.

$d(\cdot, \cdot)$ to denote Euclidean distance. The radial mask $M_r$ is defined as:

$$M_r(u, v) := \begin{cases} 1, & \text{if } d((u,v),(c_u,c_v)) \leq r \\ 0, & \text{otherwise} \end{cases} \quad (1)$$

The mask is applied on the zero-shifted output of the Fourier transform of each image, denoted $X$, followed by the inverse transform, i.e. $X_{filtered} = \mathcal{F}^{-1}(\mathcal{F}(X) \odot M_r)$, where $\odot$ is the element-wise product. Fourier-filtering is performed on each color channel independently. A standard CNN suffers up to a $75\%$ drop in accuracy on low-pass filtered CIFAR10 data as it relies on high frequency information that is no longer present (Table 1). On the other hand, the LSF-REGULARIZED model performs well on both clean images as well severely low-pass filtered images. This shows that regularized CNNs are able to exploit very low frequency features ($r \leq 5$) not easily recognizable even to humans, which reflects the limited spatial-frequency-sensitivity range of the human eye.

| Method | CIFAR10 | | | | CIFAR100 | | | |
|---|---|---|---|---|---|---|---|---|
| | clean | $r = 11$ | $r = 7$ | $r = 5$ | clean | $r = 11$ | $r = 7$ | $r = 5$ |
| Std. Train | 94.9 | 78.1 | 24.9 | 18.6 | 76.2 | 49.7 | 14.1 | 6.6 |
| LSF-REGULARIZED | 86.3 | 86.2 | **84.4** | **78.3** | 61.4 | 61.5 | **58.0** | **46.8** |
| MSF-REGULARIZED | 87.2 | 86.3 | 71.5 | 46.2 | 62.4 | 62.2 | 46.4 | 18.6 |
| ASF-REGULARIZED | 87.6 | 85.0 | 69.3 | 45.0 | 67.0 | 62.1 | 41.1 | 19.8 |
| AT (PGD $\ell_2, \epsilon = 1$) | 81.6 | 80.2 | 76.1 | 67.5 | 58.8 | 56.8 | 50.0 | 40.2 |
| Gaussian Noise | 94.5 | 84.4 | 32.4 | 19.5 | 73.1 | 61.9 | 27.7 | 11.6 |
| AugMix | 95.8 | **92.0** | 79.4 | 50.2 | 77.8 | **64.0** | 45.6 | 24.6 |

Table 1: Accuracy on Fourier-filtered CIFAR10/CIFAR100 test sets.

The MSF-REGULARIZED model performs better for mild Fourier-filtering ($r = 11$ and $r = 7$) but suffers a big drop in accuracy ($71.5\%$ to $46.2\%$ in CIFAR10 and $46.4\%$ to $18.6\%$ in CIFAR100) for strong low-pass filtering ($r = 5$). We also observed that the ASF-REGULARIZED model is also significantly more robust than the baseline model although not as robust as the LSF-REGULARIZED model to Fourier-filtering. This shows that the ASF-REGULARIZED model can use coarse low-spatial-frequency features in addition to medium and high spatial frequencies. The low-frequency sensitivity of adversarially trained models makes it more robust than the baseline model but less so than the LSF-REGULARIZED model, while the Gaussian noise augmented model suffers large drops in accuracy. AugMix tends to be robust to mild Fourier-filtering but suffers large drops in accuracy under more severe filtering ($r = 5$) due to its reliance on mid-to-high-frequency information, as suggested by its *SFS* (Figure 3), in comparison to the LSF-REGULARIZED model.

## 4.3 PERFORMANCE UNDER FOURIER-NOISE CORRUPTIONS

Recently, data-agnostic corruptions along Fourier-basis directions have been identified as a threat to model robustness and security (Yin et al., 2019; Tsuzuku & Sato, 2019). A Fourier-noise corruption is additive noise containing a single Fourier-mode (frequency). These corruptions are semantics-preserving and significantly affect model performance while being recognizable to humans, please refer to Appendix E for examples in each dataset. We generated corruptions for each of 1,024 Fourier-modes in SVHN/CIFAR10/CIFAR100 and report the overall mean accuracy across all corruptions in Table 2. We note that the baseline model suffers significant drops in accuracy across all

datasets. The LSF-regularized model achieves the highest mean accuracy in SVHN while Gaussian augmentation also improves over the baseline. AugMix (Hendrycks et al., 2020), a data-augmentation method proposed for robustness, performs worse than baseline. In CIFAR10 and CIFAR100, the MSF-regularized model achieves the highest mean accuracies. Detailed heat maps containing error rates for each Fourier-mode can be found in Appendix E.

| Method | SVHN | | | CIFAR10 | | | CIFAR100 | | |
|---|---|---|---|---|---|---|---|---|---|
| | clean | $\epsilon$=3 | $\epsilon$=4 | clean | $\epsilon$=3 | $\epsilon$=4 | clean | $\epsilon$=3 | $\epsilon$=4 |
| Std. Train | 96.4 | 81.9 | 77.4 | 94.9 | 40.8 | 31.5 | 76.2 | 22.3 | 14.9 |
| LSF-REGULARIZED | 95.1 | **92.1** | **91.0** | 86.3 | 52.4 | 47.5 | 61.4 | 33.9 | 30 |
| MSF-REGULARIZED | 93.1 | 77.1 | 70.9 | 87.2 | **62.4** | **54.3** | 62.4 | **42.6** | **37.3** |
| ASF-REGULARIZED | 96.1 | 78.3 | 71.1 | 87.6 | 60.8 | 48.7 | 67 | 21.6 | 14.6 |
| AugMix | 96.6 | 75.7 | 70.1 | 95.8 | 53.0 | 41.0 | 77.8 | 29.1 | 22.7 |
| Gaussian Noise | 96.4 | 89 | 85.1 | 94.5 | 57.2 | 40.8 | 73.1 | 33.7 | 20.9 |

Table 2: Mean accuracy across all Fourier-noise corruptions averaged across 1,024 randomly selected test samples for each corruption. $\ell_2$ norms of the additive Fourier-noise are $\epsilon \in \{3, 4\}$.

## 4.4 PERFORMANCE UNDER CORRUPTIONS IN (HENDRYCKS & DIETTERICH, 2019)

We evaluated the robustness of spatial frequency regularized models to corruptions (please see Appendix F for examples) in CIFAR10-C (Hendrycks & Dietterich, 2019). In addition, we applied these corruptions to SVHN to create SVHN-C to evaluate the performance of methods across datasets. Overall, SFS regularization provided consistent improvements across both SVHN-C (Table 3) and CIFAR10-C (Table 4) due to its general and principled approach. We observed that AugMix's (Hendrycks et al., 2020) high performance on CIFAR10-C does not translate to high performance on SVHN-C. Similarly, Gaussian augmentation's high performance in SVHN-C did not translate to CIFAR10-C. Specifically, on SVHN-C, LSF-regularization, low-pass filtering and Gaussian augmentation performed very well under noise corruptions whereas AugMix did not improve over the baseline. In CIFAR10-C, we observed that the LSF-REGULARIZED model was significantly more robust than the baseline model to all *blurring* corruptions as well as *pixelate, elastic, snow and frost* corruptions. The baseline model suffered up to 45% drop in accuracy on these corruptions, whereas the LSF-REGULARIZED model suffered at most 5% drop in accuracy. This is likely due to the robustness of low-spatial-frequency features to corruptions such as blurring. We also observed that the MSF-REGULARIZED and ASF-REGULARIZED models were more robust to *noise* corruptions as well as JPEG compression (Table 4), which suggests multiple spatial frequency bands maybe needed to do well under such corruptions. We observed similar results in CIFAR100-C (Appendix F.2). Overall, SFS regularization can provide consistent improvements in robustness to these corruptions across datasets due to its general and principled approach. On the other hand, data augmentation based methods invariably perform better on some datasets than others depending on the statistics and features of each dataset.

| Method | Clean | Noise | | | Blur | | | | Weather | | | | Digital | | | |
|---|---|---|---|---|---|---|---|---|---|---|---|---|---|---|---|---|
| | | Gauss. | Shot | Impulse | Defocus | Glass | Motion | Zoom | Snow | Frost | Fog | Bright | Contrast | Elastic | Pixel | JPEG |
| Std. Train | 96.4 | 81.3 | 84.4 | 72.7 | 95.9 | 92.4 | 94.9 | 96.1 | 75.5 | 84.0 | 48.9 | 94.5 | 89.4 | 90.0 | 95.8 | 95.9 |
| LSF-REGULARIZED | 95.1 | 90.8 | 91.6 | 84.7 | 94.6 | 92.9 | 93.5 | 95.2 | 82.4 | 88.4 | 42.1 | 93.2 | 83.6 | 91.1 | 94.8 | 94.9 |
| MSF-REGULARIZED | 93.1 | 68.3 | 71.8 | 52.6 | 92.7 | 82.8 | 90.9 | 93.6 | 75.2 | 76.6 | 40.7 | 90.7 | 62.8 | 84.1 | 91.8 | 92.2 |
| ASF-REGULARIZED | 96.1 | 80.9 | 83.3 | 73.2 | 95.5 | 91.7 | 94.4 | 96.0 | 79.2 | 84.3 | 45.5 | 94.3 | 85.7 | 91.0 | 95.2 | 95.7 |
| Low-pass filtered | 95 | 90.5 | 91.4 | 85.1 | 94.7 | 93.0 | 93.7 | 95.1 | 82.9 | 88.1 | 40.8 | 92.7 | 85.8 | 90.5 | 94.8 | 94.8 |
| AT (PGD $\ell_\infty$, 8/255) | 96.9 | 85.6 | 88.1 | 73.8 | 96.0 | 94.2 | 95.0 | 96.6 | 80.1 | 88.8 | 43.9 | 95.0 | 72.2 | 93.2 | 96.5 | 96.6 |
| Gaussian Noise | 96.4 | 92.8 | 93.8 | 84.4 | 95.5 | 94.4 | 94.4 | 96.2 | 80.7 | 89.3 | 42.6 | 94.4 | 74.4 | 92.4 | 96.0 | 96.1 |
| AugMix | 96.6 | 79.1 | 82.5 | 74.7 | 96.4 | 92.1 | 95.4 | 96.4 | 73.4 | 83.1 | 50.0 | 95.2 | 93.5 | 90.5 | 95.2 | 96.2 |

Table 3: Accuracies on SVHN-C corruptions of highest severity level. We created SVHN-C by applying corruptions in CIFAR10-C to SVHN test samples.

| Method | Clean | Noise | | | Blur | | | | Weather | | | | Digital | | | |
|---|---|---|---|---|---|---|---|---|---|---|---|---|---|---|---|---|
| | | Gauss. | Shot | Impulse | Defocus | Glass | Motion | Zoom | Snow | Frost | Fog | Bright | Contrast | Elastic | Pixel | JPEG |
| Std. Train | 94.9 | 31.5 | 37.9 | 33.8 | 54.9 | 49.5 | 67.3 | 65.2 | 75.8 | 65.4 | 74.2 | 92.0 | 56.3 | 73.2 | 41.0 | 73.5 |
| LSF-REGULARIZED | 86.3 | 29.4 | 32.5 | 20.0 | 83.9 | 81.8 | 80.4 | 83.9 | 79.6 | 79.2 | 60.6 | 82.0 | 28.3 | 82.5 | 85.5 | 65.3 |
| MSF-REGULARIZED | 87.2 | 55.6 | 59.2 | 34.9 | 78.0 | 71.2 | 67.7 | 77.5 | 77.8 | 78.4 | 60.4 | 82.8 | 19.5 | 76.1 | 83.5 | 81.4 |
| ASF-REGULARIZED | 87.6 | 62.3 | 64.6 | 44.6 | 71.3 | 70.2 | 69.7 | 75.3 | 75.9 | 74.6 | 47.3 | 83.6 | 26.5 | 76.1 | 79.7 | 82.6 |
| Low-pass filtered | 86.6 | 58.0 | 60.4 | 43.5 | 86.0 | 84.0 | 82.5 | 85.5 | 81.0 | 80.3 | 62.7 | 82.5 | 35.2 | 83.5 | 86.0 | 71.6 |
| AT (PGD $\ell_2, \epsilon = 1$) | 81.6 | 74.6 | 75.5 | 66.7 | 74.9 | 74.2 | 73.2 | 76.5 | 75.2 | 70.5 | 35.1 | 77.9 | 19.3 | 75.8 | 78.4 | 79.7 |
| Gaussian Noise | 94.5 | 44.5 | 53.6 | 29.4 | 53.6 | 55.4 | 62.6 | 66.3 | 80.6 | 78.0 | 68.6 | 91.6 | 27.4 | 73.8 | 54.5 | 82.2 |
| AugMix | 95.8 | 66.1 | 72.7 | 75.6 | 92.6 | 75.0 | 90.8 | 91.6 | 86.6 | 85.4 | 83.2 | 93.6 | 86.6 | 85.4 | 83.2 | 93.6 |

Table 4: Accuracies on CIFAR10-C corruptions of highest severity level.

### 4.5 EVALUATION OF GLOBAL FEATURE LEARNING

Here we wish to evaluate the extent to which models use global features by measuring their performance on patch-shuffled images, which have previously been used by Mummadi et al. (2021); Zhang & Zhu (2019); Wang et al. (2019). Patch-shuffling involves splitting an image into $k \times k$ squares and randomly swapping the positions of these squares. This is intended to destroy global features and retain local features with larger values of $k$ retaining less global structure in the image (please see Appendix G for examples). As such, models that rely more on global rather than local structure suffer more from patch-shuffling. Hence, in this benchmark, lower accuracy suggests increased reliance on and use of global structure. We observed that spatial frequency regularized models, adversarially trained models suffered larger drops in accuracy compared to the standard trained, AugMix and Gaussian augmentation models, which suggests their reliance on global structure in images contributes to their robustness (Table 5).

| Method | CIFAR10 | | | CIFAR100 | | |
|---|---|---|---|---|---|---|
| | clean | $k = 2$ | $k = 3$ | clean | $k = 2$ | $k = 3$ |
| Std. Train | 94.9 | 66.5 | 45.8 | 76.2 | 39.9 | 21.4 |
| AugMix | 95.8 | 67.3 | 48.7 | 77.8 | 40.7 | 23.8 |
| Gaussian Noise | 94.5 | 62.9 | 44.5 | 73.1 | 34.4 | 18 |
| LSF-REGULARIZED | 86.3 | **43.2** | **30.6** | 61.4 | **23.4** | **13** |
| MSF-REGULARIZED | 87.2 | **46.8** | **33.1** | 62.4 | **24.1** | **13.0** |
| ASF-REGULARIZED | 87.6 | **46.8** | **32.6** | 67 | **29.0** | **15** |
| Low-pass filtered | 86.6 | **43.4** | **30.0** | 60 | **20.4** | **10.4** |
| AT (PGD $\ell_2, \epsilon = 1$) | 81.6 | **45.2** | **35.0** | 58.8 | **19.1** | **11.1** |

Table 5: Accuracy on patch-shuffled CIFAR10/CIFAR100 test sets.

## 5 CONCLUSION

We proposed a novel and rigorous measure of the *spatial frequency sensitivity (SFS)* of any differentiable computer model using its input-Jacobian represented in the Fourier-basis. We showed that models display consistent *SFS* across samples as well as across architectures for a given task and when there is a shift in the Fourier-statistics of features relative to a model's frequency sensitivity, performance can suffer. We also proposed a novel method of spatial frequency regularization that can be used to improve robustness to many test-time corruptions. Extending our method to other computer vision tasks (e.g. detection) as well as domains ($n$-dimensional) are interesting directions for future work.

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

APPENDICES

# A  SPATIAL FREQUENCY SENSITIVITY AND REGULARIZER

## A.1  PSEUDO-CODES

---

**Algorithm 1:** Spatial Frequency Sensitivity (SFS)

---

**Input:** Labeled samples $\mathbf{L} = \{(x_i, y_i)\}_{i=1}^{n}$; a model $f$ with trained parameters $\theta$
**Output:** Estimated Spatial Frequency Sensitivity of model, $f_{SFS}(\cdot; \theta)$

**for** $i = 1, \ldots, n$ **do**
    compute loss $\mathcal{L}_{\text{CE}}(f(x_i), y_i)$ // `forward pass`
    backpropagate $\mathcal{L}_{\text{CE}}$ to obtain $\frac{\partial \mathcal{L}_{\text{CE}}}{\partial x_i}$ // `input-jacobian`
    $\frac{\partial \mathcal{L}_{\text{CE}}}{\partial x_{f_i}} = \mathcal{F}(\frac{\partial \mathcal{L}_{\text{CE}}}{\partial x_i})$ // `2D DFT of input-jacobian (averaged across`
    `color channels)`
    $f_{SFS}(x_i, y_i) = [P_k; \text{for k=1 to N/}\sqrt{2}]$ // `P`$_k$ `is proportion of total power`
    `(excluding DC) in coefficients at radial distance` $k$
**end**
$f_{SFS}(\cdot; \theta) = \frac{1}{n} \sum_{i=1}^{n} f_{SFS}(x_i, y_i)$ // `estimated SFS of model`

---

**Algorithm 2:** Spatial Frequency Regularized Minibatch Training

---

**Input:** Labeled samples $\mathbf{L} = \{(x_i, y_i)\}_{i=1}^{n}$; a model $f$ with parameters $\theta$; SFS $\in$
    $\{LSF, MSF, ASF\}$; hyperparameter $\lambda_{\text{SFS}}$
**Output:** Regularized model with updated parameters $\theta$

**for** $i = 1, \ldots, n$ **do**
    compute loss $\mathcal{L}_{\text{CE}}(f(x_i), y_i)$ // `forward pass`
    $\mathcal{L}(x_i, y_i) = \mathcal{L}_{\text{CE}}(x_i, y_i) + \lambda_{\text{SFS}} \mathcal{L}_{\text{SFS}}(x_i, y_i)$ // `backpropagate` $\mathcal{L}$ `to update` $\theta$
**end**

---

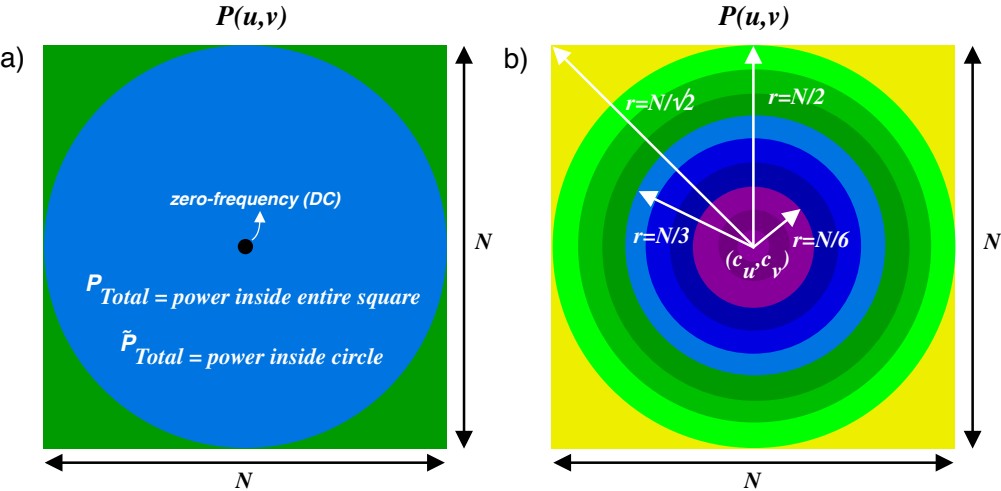

Figure 5: a) Illustration of the power-matrix $P$ of the Fourier-transformed input-Jacobian. $P_{Total}$, the power in all components excluding the zero-frequency component is used to normalize the proposed $SFS$ of the model at a particular input. b) LSF (purple), MSF (blue) and ASF (purple, blue and green) frequency bands used for spatial frequency regularization. Best viewed in color.

## B    SPATIAL FREQUENCY CHANNELS IN THE BRAIN

To further motivate the spatial frequency perspective of visual representation learning, we briefly describe some relevant findings from neuroscience and vision research. While the brain does not strictly perform a Fourier analysis of visual scenes, there has been mounting evidence over decades for multiple spatial frequency channels in the early human visual system that are physiologically independent and selectively responsive to distinct spatial frequency bands (Campbell & Robson, 1964; 1968). It has been posited that the use of different spatial frequency channels are determined by the demands of a given visual task through an attention mechanism (Schyns & Oliva, 1999; Rotshtein et al., 2010; Julesz & Papathomas, 1984). Spatial frequency channels enable us to attend to different spatial scales in a scene at a fixed viewing distance, similar to the focus lens in a camera (Figure 6). On the other hand, once trained, a CNN's *spatial frequency sensitivity* (Section 3.1) is inflexible and does not vary much across samples (both in and out of its training data distribution).

```
EFEFEFEFEFEFEFEFEFEFEFEFEFEFEFEFEFEFEFEF
EFEFEFEFEFEFEFEFEFEFEFEFEFEFEFEFEFEFEFEF
EFEFEFEFEFEFEFEFEFEFEFEFEFEFEFEFEFEFEFEF
EFEFEFEFEFEFEFEFEFEFEFEFEFEFEFEFEFEFEFEF
EFEFEFEFEFEFEFEFEFEFEFEFEFEFEFEFEFEFEFEF
EFEFEFEFEFEFEFEFEFEFEFEFEFEFEFEFEFEFEFEF
EFEFEFEFEFononononononononononononoEFEFEFEFEF
EFEFEFEFEFonononononononononononononoEFEFEFEFEF
EFEFEFEFEFononononononononononononononoEFEFEFEF
EFEFEFEFEFonononononononononononononoEFEFEFEFEF
EFEFEFEFEFononononononononononononoEFEFEFEFEF
EFEFEFEFEFEFEFEFEFEFEonononononEFEFEFEFEFEFEFEFEFE
EFEFEFEFEFEFEFEFEFEFEonononononEFEFEFEFEFEFEFEFEFE
EFEFEFEFEFEFEFEFEFEFEonononononEFEFEFEFEFEFEFEFEFE
EFEFEFEFEFEFEFEFEFEFEonononononEFEFEFEFEFEFEFEFEFE
EFEFEFEFEFEFEFEFEFEFEonononononEFEFEFEFEFEFEFEFEFE
EFEFEFEFEFEFEFEFEFEFEonononononEFEFEFEFEFEFEFEFEFE
EFEFEFEFEFEFEFEFEFEFEonononononEFEFEFEFEFEFEFEFEFE
EFEFEFEFEFEFEFEFEFEFEonononononEFEFEFEFEFEFEFEFEFE
EFEFEFEFEFEFEFEFEFEFEonononononEFEFEFEFEFEFEFEFEFE
EFEFEFEFEFEFEFEFEFEFEonononononEFEFEFEFEFEFEFEFEFE
EFEFEFEFEFEFEFEFEFEFEonononononEFEFEFEFEFEFEFEFEFE
EFEFEFEFEFEFEFEFEFEFEonononononEFEFEFEFEFEFEFEFEFE
EFEFEFEFEFEFEFEFEFEFEonononononEFEFEFEFEFEFEFEFEFE
EFEFEFEFEFEFEFEFEFEFEonononononEFEFEFEFEFEFEFEFEFE
EFEFEFEFEFEFEFEFEFEFEFEFEFEFEFEFEFEFEFEFEFEFEFEF
EFEFEFEFEFEFEFEFEFEFEFEFEFEFEFEFEFEFEFEFEFEFEFEF
EFEFEFEFEFEFEFEFEFEFEFEFEFEFEFEFEFEFEFEFEFEFEFEF
EFEFEFEFEFEFEFEFEFEFEFEFEFEFEFEFEFEFEFEFEFEFEFEF
EFEFEFEFEFEFEFEFEFEFEFEFEFEFEFEFEFEFEFEFEFEFEF
```

Figure 6: Letters at multiple spatial scales. This image comprises the letters o, n, E and F at a small spatial scale (HSF). The letter T is also visible at a larger spatial scale (LSF) formed by the specific arrangement of the letters o, n amidst letters E, F. Identifying these letters requires processing features at multiple scales, enabled by distinct spatial frequency channels in the early visual cortex. Our ability to recognize only one of these scales at a time is evidence for the physiological independence of spatial frequency channels in the brain. Image based on Julesz & Papathomas (1984).

# C  SUPPLEMENTARY PLOTS

## C.1  SPATIAL FREQUENCY SENSITIVITIES OF POPULAR IMAGENET MODELS

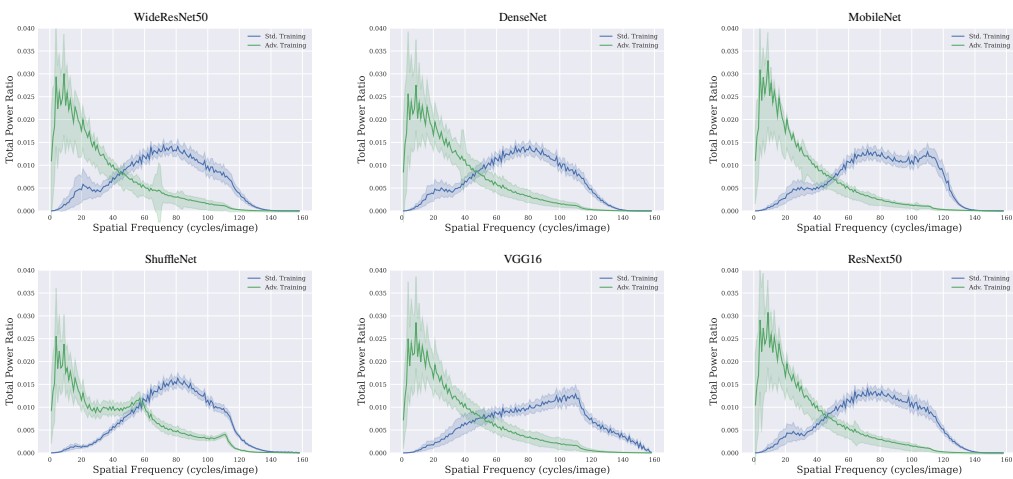

Figure 7: Spatial frequency sensitivities of popular ImageNet architectures after standard training and adversarial training using PGD-$\ell_2$ ($\epsilon = 3$) attacks. The *SFS* is consistent across architectures for both standard and adversarially trained models although VGG16 shows increased sensitivity to high frequencies. The shaded region represents two standard deviations.

## C.2  SPATIAL FREQUENCY SENSITIVITIES OF MODELS TRAINED ON CIFAR10 AND CIFAR100

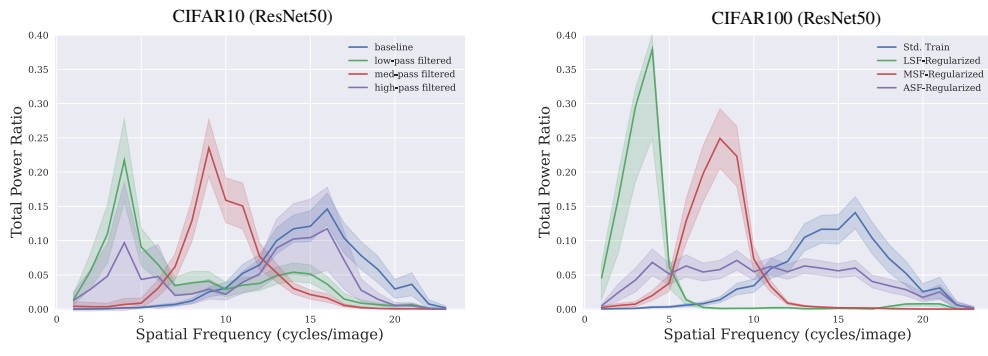

Figure 8: Various methods trained on CIFAR10 (left) and CIFAR100 (right). The shaded region represents two standard deviations.

## C.3  SPATIAL FREQUENCY SENSITIVITIES OF MODELS TRAINED ON SVHN

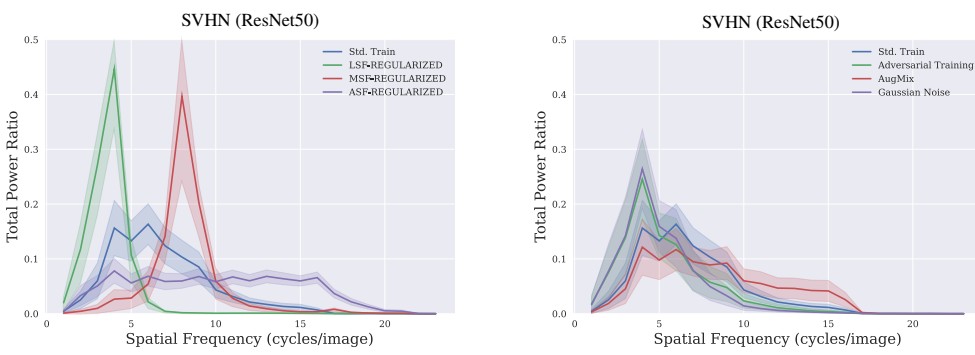

Figure 9: Various methods trained on SVHN. The shaded region represents two standard deviations.

## C.4  TRAINING ON CIFAR10 CORRUPTIONS

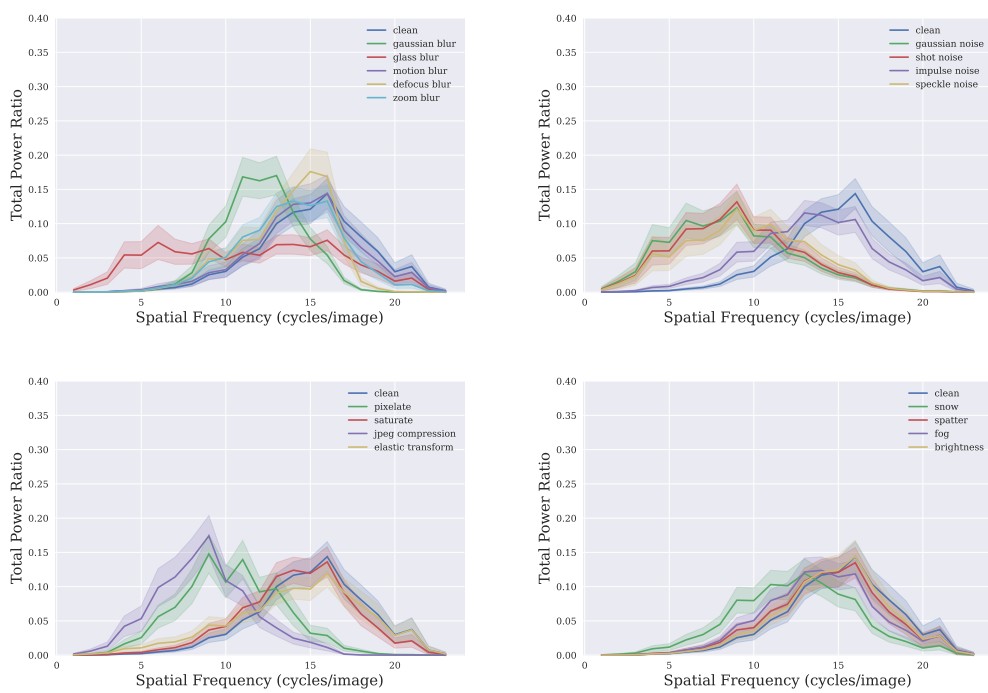

Figure 10: *Spatial frequency sensitivities* of ResNet50 models trained on the CIFAR10 training set distorted by corruptions derived from the CIFAR10-C (severity 5) dataset. The shaded region represents two standard deviations.

# D FOURIER FILTERING

## D.1 RADIAL FILTERING

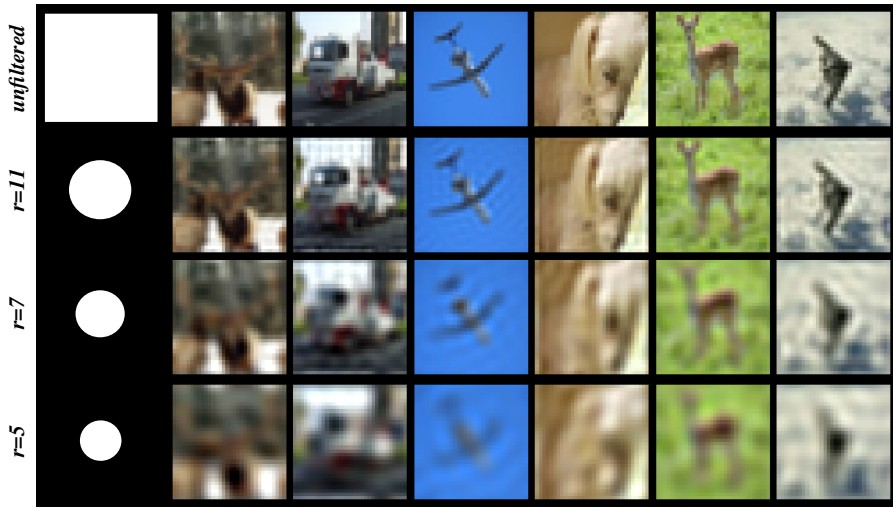

Figure 11: First image in each row is the mask in Fourier space (lowest frequency at centre). White pixels preserve and black pixels set Fourier components to zero. Top row are original CIFAR10 images, other rows are Fourier-filtered with different radial masks.

## D.2 BAND-PASS FILTERING

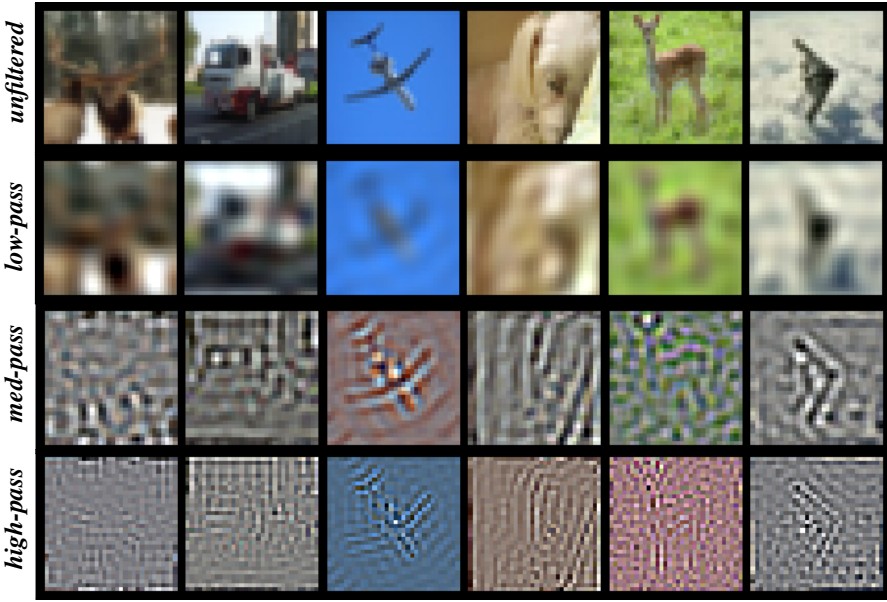

Figure 12: For band-pass Fourier-filtering CIFAR10 training images, we filtered Fourier-coefficients in each color channel separately. For low-pass filtering, Fourier-coefficients with radial distance $r(u,v) > 5$ were set to zero. For medium-pass filtering, Fourier-coefficients with $r(u,v) < 5$ and $r(u,v) > 10$ were set to zero. In high-pass filtering, Fourier-coefficients with $r(u,v) < 10$ were set to zero. Images contast-maximised for viewing.

### D.3 SPATIAL FREQUENCY REGULARIZATION IS NOT EQUIVALENT TO TRAINING ON FOURIER-FILTERED DATA

We note that spatial frequency regularization is not equivalent to standard training on corresponding Fourier-filtered data. In natural images, the amount of energy in frequency bands falls off rapidly as frequency increases (Hyvärinen et al., 2009), hence, medium and high-pass filtered natural images typically appear completely empty to the human eye without additional contrast maximisation. Even still, such images are not easily recognizable to the human eye due to its and reliance on low frequency information to recognize objects optimally (Appendix D.2). We observed that standard training of CNNs on medium or high-pass Fourier-filtered data did not achieve high clean accuracies (Tables 4, 6). A model trained on medium-pass Fourier-filtered CIFAR10 achieved a clean accuracy of only $\sim$33% on clean test samples whereas the MSF-REGULARIZED model's clean accuracy is $\sim$87%, and a model trained on high-pass filtered CIFAR10 training samples achieved only $\sim$15% accuracy on clean test samples. Due to the energy statistics over frequency bands in natural images, training on Fourier-filtered data is not successful for all but the lowest frequency bands, where most of their energy resides. In addition, we note that regularizers such as ASF-regularization cannot be recreated by Fourier-filtering the training data due to the tendency of standard training to be sensitive to certain frequencies.

# E  BLACK BOX FOURIER-NOISE CORRUPTIONS

## E.1  CIFAR10 FOURIER-NOISE CORRUPTIONS

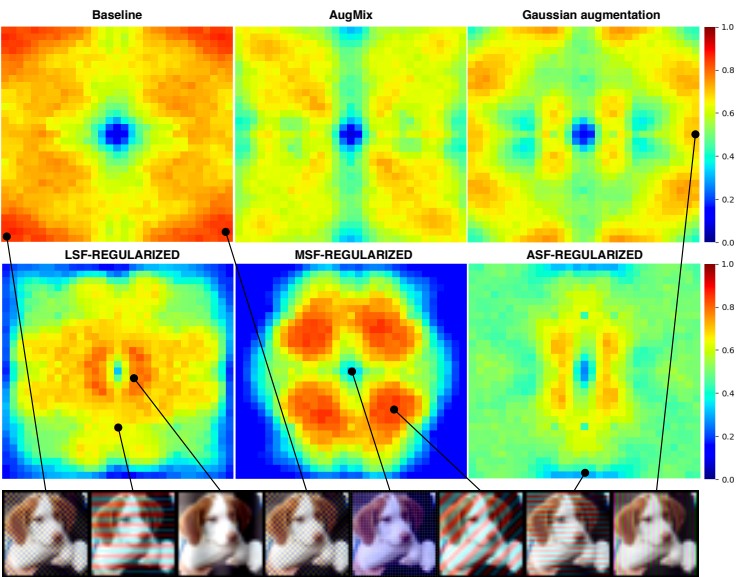

Figure 13: (CIFAR10) Heat map of error rates for each Fourier-mode corruption. Each pixel in the heat map is the error of the model when the corresponding Fourier-mode noise ($\epsilon = 4$) is added to the inputs. The bottom row displays example images containing the corresponding Fourier-corruption.

## E.2  SVHN FOURIER-NOISE CORRUPTIONS

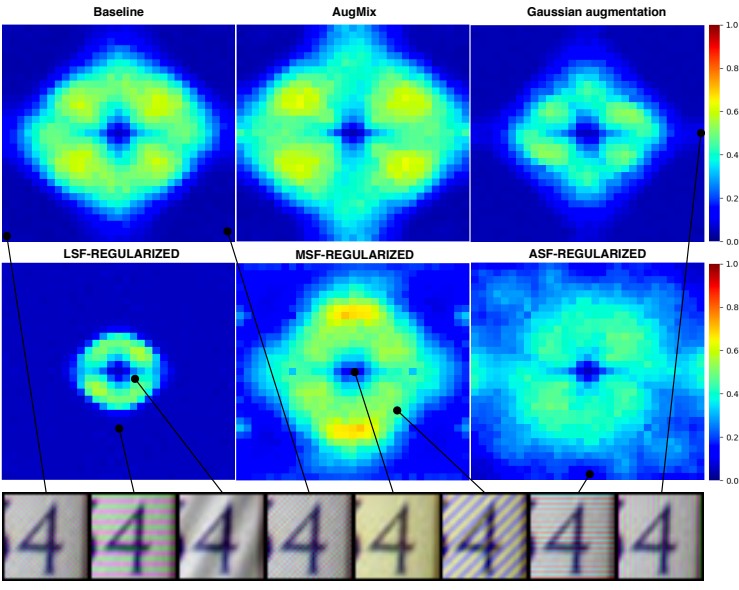

Figure 14: (SVHN) Heat map of error rates for each Fourier-mode corruption. Each pixel in the heat map is the error of the model when the corresponding Fourier-mode noise ($\epsilon = 4$) is added to the inputs. The bottom row displays example images containing the corresponding Fourier-corruptions.

# F CORRUPTIONS IN (HENDRYCKS & DIETTERICH, 2019)

## F.1 EXAMPLES

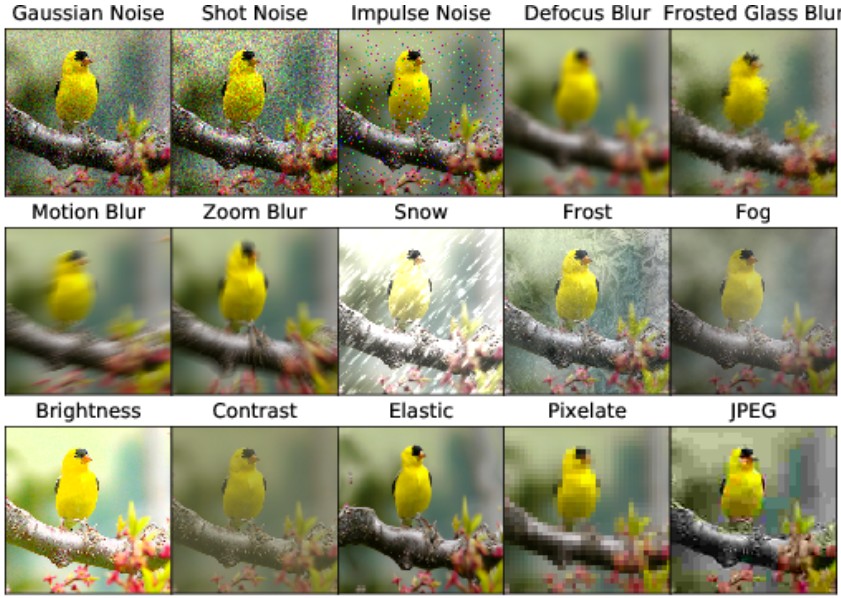

Figure 15: Examples of image corruptions curated by Hendrycks & Dietterich (2019) to evaluate robustness of vision models.

## F.2 CIFAR100-C BENCHMARK

| Method | Clean | Noise | | | Blur | | | | Weather | | | | Digital | | | |
|---|---|---|---|---|---|---|---|---|---|---|---|---|---|---|---|---|
| | | Gauss. | Shot | Impulse | Defocus | Glass | Motion | Zoom | Snow | Frost | Fog | Bright | Contrast | Elastic | Pixel | JPEG |
| Std. Train | 76.2 | 10.9 | 12.5 | 8.7 | 30.5 | 18.3 | 41.1 | 37.7 | 46.0 | 30.6 | 40.0 | 69.9 | 29.8 | 49.6 | 21.4 | 48.9 |
| LSF-REGULARIZED | 61.4 | 16.0 | 18.8 | 8.4 | 56.5 | 56.3 | 52.9 | 57.9 | 52.0 | 51.1 | 29.2 | 54.2 | 13.6 | 56.1 | 60.0 | 36.3 |
| MSF-REGULARIZED | 62.4 | 31.8 | 34.6 | 16.2 | 49.6 | 50.1 | 44.6 | 52.9 | 51.6 | 48.7 | 28.0 | 54.3 | 9.2 | 53.4 | 58.7 | 54.5 |
| ASF-REGULARIZED | 67.0 | 15.6 | 18.4 | 8.6 | 41.4 | 40.9 | 43.0 | 47.0 | 51.4 | 44.6 | 26.9 | 58.9 | 11.8 | 52.6 | 61.0 | 54.2 |
| Low-pass filtered | 60.1 | 32.7 | 34.9 | 18.9 | 59.3 | 56.6 | 54.8 | 58.1 | 50.2 | 49.4 | 27.0 | 54.1 | 13.4 | 57.0 | 59.9 | 43.8 |
| AT (PGD $\ell_2, \epsilon = 1$) | 58.8 | 44.8 | 45.9 | 26.0 | 48.8 | 49.4 | 46.3 | 50.0 | 46.6 | 40.9 | 10.4 | 50.2 | 5.0 | 50.7 | 54.7 | 55.5 |
| Gaussian noise | 73.1 | 25.8 | 30.8 | 12.8 | 32.9 | 33.5 | 36.1 | 39.9 | 55.4 | 51.4 | 30.7 | 66.6 | 11.9 | 52.3 | 41.2 | 59.0 |
| AugMix | 77.8 | 36.8 | 41.9 | 52.9 | 71.8 | 47.8 | 69.4 | 70.3 | 60.3 | 55.7 | 47.7 | 70.0 | 47.4 | 61.0 | 50.0 | 60.9 |

Table 6: Accuracies for CIFAR100-C corruptions of highest severity level. We note that the LSF-REGULARIZED model is significantly more robust than the baseline for blurring corruptions. For noise corruptions, the MSF-REGULARIZED model is more robust compared to the baseline as well as the LSF-REGULARIZED model. Similar to CIFAR10-C, low-pass filtering and AT also lead to models robust to blurring and noise. MSF-REGULARIZED and ASF-REGULARIZED models are again more robust to noise and JPEG noise, compared to the ASF-REGULARIZED model.

## G   PATCH-SHUFFLING IMAGES

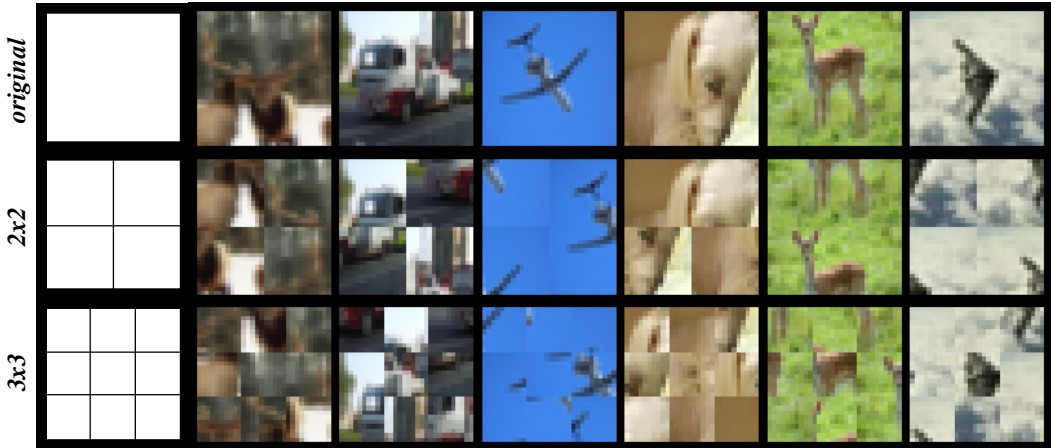

Figure 16: Patch-shuffling: Images are partitioned into squares whose positions are randomly exchanged. This operation destroys global structure in the image and is used to evaluate the extent to which a model relies on global information.

