# OpenReview forum: "Spatial Frequency Sensitivity Regularization for Robustness"
_ICLR.cc/2022/Conference — ICLR 2022 Submitted_

### Official Review · Reviewer_fjLz · 2021-11-02

**Correctness:** 4
**Technical Novelty And Significance:** 3
**Empirical Novelty And Significance:** 3
**Recommendation:** 6
**Confidence:** 3

**Main Review:**

Strength

1) This paper is clearly written and easy to follow.
2) The proposed SFS regularizer is clearly motivated. It is more interpretable than existing sensitivity regularization techniques (Section 2.1). It is also generic to be applied to any differentiable model.
3) The experiments show that the proposed SFS regularizer is robust to Fourier filtering and corruptions, while maintains high accuracy on clean images.

Weakness

1) The experiment datasets and models look insufficient to me. The authors only show results on CIFAR10/100, but other baseline methods (e.g., AugMix) show at least some ImageNet results.
2) The presented results are bit unconvincing to conclude the method’s effectiveness. In Table 2 and 3, the SFS regularization methods only match with the performance of the “Low-pass filtered" method, and their performances highly correlate through all types of corruptions. Although the authors point out their technical difference (Section 4.6), I still doubt the advantage of SFS regularization over the simple "Low-pass filtered" method regarding the current performance.

Questions

1) What is the computational cost compared with other baseline methods, considering the regularization term involves Jacobian?

**Summary Of The Paper:**

This paper proposes a measure for a model’s spatial frequency sensitivity (SFS) based on the input-Jacobian in the Fourier basis. With this measure, the authors observe standard CNN training biases towards certain particular spatial frequencies consistently across samples. Based on this measure, the authors propose a family of spatial frequency regularization techniques to suppress the model’s sensitivities to certain spatial frequencies.


**Summary Of The Review:**

The proposed SFS measures are clearly motivated, and provide interesting insights from the frequency-perspective on how existing methods improves model’s robustness. My major concern lies in the insufficient experiments and unconvincing results as stated in the weakness section.

**Disclaimer**: I am unfamiliar with robustness in CV, it's possible I have some misunderstandings. My rating is highly subject to change based on other reviewers’ comments.

---

> ### Author Response · Authors · 2021-11-23
> **Response**
>
> Dear reviewer,
>
> Thanks for your valuable feedback and comments.
>
> > What is the computational cost compared with other baseline methods, considering the regularization term involves Jacobian?
>
> We briefly describe the computations involved in each approach:
>
> a) The proposed Jacobian-regularizer requires an additional backprop step to compute the SFS-loss followed by computing derivatives again through this backward graph. The Fourier transform computation is highly efficient thanks to FFT available in libraries such as PyTorch. The frequency band indices in the DFT matrix can be pre-computed for fast access.
>
> b) AugMix requires generating and mixing multiple (min: 3 and max: 9, with default settings) augmentations in addition to computing the Jenson-Shannon divergence loss term to penalize posterior difference between augmented images.
>
> c) Adversarial training (n-step PGD) requires n additional forward-backward passes in order to iteratively find a strong adversarial perturbation using the input-Jacobian.
>
> d) Gaussian noise augmentation requires generating i.i.d. additive gaussian noise to each pixel.
>
> > I still doubt the advantage of SFS regularization over the simple "Low-pass filtered" method regarding the current performance
>
> While low-pass filtering may be a good alternative to LSF-regularization, medium-pass and high-pass filtering do not lead to successful training (we discuss this in Appendix D.3, previously Section 4.6). MSF-regularization has better performance in some settings such as JPEG compression noise (Table 3, last column) as well as low or high-frequency Fourier-distortions (Section 4.3). In addition, there is no Fourier-filter that exactly reproduces ASF-regularization.
>
> > The authors only show results on CIFAR10/100, but other baseline methods (e.g., AugMix) show at least some ImageNet results”
>
> We were unable to add a large-scale benchmark on ImageNet due to limited resource. We have added an additional benchmark (Section 4.3) including the SVHN 	dataset and also added SVHN-C to Section 4.4 to improve the diversity of benchmarks. We show that SFS regularization provides improved robustness in new datasets and distortions due to its generality and principled approach.

---

### Official Review · Reviewer_womq · 2021-11-02

**Correctness:** 3
**Technical Novelty And Significance:** 2
**Empirical Novelty And Significance:** 2
**Recommendation:** 5
**Confidence:** 4

**Main Review:**

**Strengths**

This paper presents several interesting discussions, in particular, I think the idea of chain rule can be directly applied to the input-Jacobian is a very neat observation.


**Weakness**

On the other hand, I think the paper can be improved on several perspectives, mostly related to how the contents can be presented and discussed.

  - Most importantly, the empirical results are presented in a non-standard way. The bold numbers are used to report performances higher than baseline in this paper, while bold numbers are mostly used to denote the highest performances.

    - By reading the numbers, I feel like the new method introduced by this method actually shows a fair strength in comparison to other methods except for AugMix. In this case, the authors might be able to argue that their method is good enough since AugMix is a very computation-heavy method specially designed on the benchmarks used in the paper (with evidence that AugMix is not that good for Table 1), but the authors seem to prefer to present the paper in a way that they seemingly want to compare to AugMix where AugMix is designed with their own method inspired by something else. As the result, the empirical results seem weak.

 - The presentation of the main idea can also be improved. For example, some illustrations on how P corresponds to the elements in Figure 1 could help understand the paper better.



**Summary Of The Paper:**

The paper is motivated by the challenge of the model's uncontrolled behavior in learning different frequency components of the image data and introduces a new regularization to improve the empirical performances. The new idea, especially the discussion where chain rule can be directly applied, is very interesting.

**Summary Of The Review:**

This paper introduces a new idea and demonstrates the empirical strength of the method, while the method has some empirical virtue, the way the results are presented makes it very hard to appreciate the results.

---

> ### Author Response · Authors · 2021-11-23
> **Response**
>
> Dear reviewer,
>
> Thanks for your valuable feedback and comments.
>
> > while the method has some empirical virtue, the way the results are presented makes it very hard to appreciate the results.
>
> We hope the updates to the overall presentation make it easier to appreciate the 	results. We have also added new benchmarks that highlight the generality and advantage of our approach over methods like AugMix through evaluations in more settings.
>
> > the empirical results are presented in a non-standard way. The bold numbers are used to report performances higher than baseline in this paper
>
> We have removed the non-standard boldening.
>
> > For example, some illustrations on how P corresponds to the elements in Figure 1 could help understand the paper better.
>
> We hope Fig 4. in Appendix A. makes this clearer to readers. We have highlighted 	the components of the power matrix P that carry different frequencies (LSF, MSF 	etc.) using colors based on radial distance from the center of the matrix (Fig 4.b).

---

### Official Review · Reviewer_GEyu · 2021-11-03

**Correctness:** 2
**Technical Novelty And Significance:** 2
**Empirical Novelty And Significance:** 1
**Recommendation:** 3
**Confidence:** 5

**Main Review:**

Major comments

The paper claims that “networks trained with our proposed regularizers obtain
significantly improved classification accuracy while maintaining high accuracy on
in-distribution clean test images.” However, all the regularization methods presented have a clean-accuracy drop of >=7.3% which is very significant, especially on a small dataset like CIFAR. This makes it hard to believe the papers claims that its method is competitive to sota-trained methods (it performs worse than AugMix, for instance). Even training with gaussian augmentation can similarly-high robustness if allowed to perform such a huge IID accuracy drop [2].

Additionally, the paper mainly tests on small datasets (CIFAR-10) and only uses Imagenet when measuring model sensitivity (never training with the regularizer). Even in this constrained setting, Fig 1 does not show gaussian-trained [1] (or patch gaussian [2] trained) network for imagenet. This lowers my confidence that the method has practical applicability.

This would not be a concern if the method was a useful tool for drawing even more insights into how models work, but this is not fleshed out in the experiments.

[1] https://papers.nips.cc/paper/2019/file/b05b57f6add810d3b7490866d74c0053-Paper.pdf
[2] https://arxiv.org/abs/1906.02611

Minor comments

It would be useful if the authors could validate the sensitivity-measuring methodology by checking that models that have been Low/Medium/High-pass filtered indeed display the sensitivities being measured.


**Summary Of The Paper:**

This paper proposes a measure of “spatial frequency sensitivity”, where a model’s input-jacobian is fourier-transformed and frequency bands are aggregated to define how sensitive a trained model is to specific fourier frequencies. The authors show this can also be used to explicitly encourage specific frequency-sensitivities, which they call “spatial frequency regularization”.


**Summary Of The Review:**

The usage of input-jacobian-fourier (which is highest input sensitivity) to measure spatial frequency sensitivity is intuitive and well explained. However, the paper neither justifies its practical applicability, nor does it develop the methods into a fleshed out model analysis to yield insightful conclusions.

---

> ### Author Response · Authors · 2021-11-23
> **Response**
>
> Dear reviewer,
>
> Thanks for your valuable feedback and comments.
>
> > It would be useful if the authors could validate the sensitivity-measuring methodology by checking that models that have been Low/Medium/High-pass filtered indeed display the sensitivities being measured
>
> We have shown this is indeed true in Fig 7. in Appendix C.2. (left).
>
> > Additionally, the paper mainly tests on small datasets (CIFAR-10) and only uses Imagenet when measuring model sensitivity
>
> Due to our limited resources, we were not able to train models on ImageNet within 	this response period. We have however added two new benchmarks to provide further perspective on the merits of our SFS approach. First, we added a new benchmark (Section 4.3) on data-agnostic distortions, which also includes results on SVHN. Second, we added SVHN-C to Section 4.4 to increase the diversity of results. Through these benchmarks, we show that SFS regularization can help improve robustness in diverse settings due to its generality and principled approach.
>
> > all the regularization methods presented have a clean-accuracy drop of >=7.3% which is very significant ... Even training with gaussian augmentation can similarly-high robustness if allowed to perform such a huge IID accuracy drop
>
> In accordance to the reviewer’s suggestion, we trained a model on CIFAR10 using Gaussian noise augmentation with mean=0. and std-dev=0.6 (we used std-dev=0.1 in the paper) that achieved 86.9% clean accuracy, which is comparable to the clean acc. of our regularized models. However, while this improved performance on Gaussian and other noise corruptions due to their statistical similarity, it did not improve performance on most other corruptions and in fact significantly degraded performance on some corruptions; accuracy on fog corruption dropped to 28.8% (from 68.6%), and to 14.3% (from 27%) for contrast corruptions. We note that in expectation i.i.d Gaussian noise has equal power across all frequencies i.e., white noise, and in principle does not provide the same level of flexibility that SFS regularization does.
>
> While we agree that our regularized models suffer drops in i.i.d. accuracy, we were more interested in the o.o.d performance of the models as that is one of the main challenges for deep learning models today. As we have seen, high i.i.d accuracy does not necessarily result in high o.o.d accuracy. We have demonstrated a new method that can significantly improve generalization performance on certain tasks and may be useful in practical applications, where i.i.d performance is not always of significant interest.

---

> > ### Comment · Reviewer_GEyu · 2021-11-29
> > **Response to Authors**
> >
> > I thank the authors for the response to my review, and for their time running additional experiments.
> >
> > My main concern, that clean-accuracy drop of >=7.3% is very significant, still remains. Without understanding why this drop happens, it's hard to assess whether the method is useful, or wether it just trades off clean accuracy for accuracy on the robustness benchmark being evaluated. For instance, as the authors point out, when training with gaussian noise, corruption accuracy is mostly improved in noise-corruptions. This means that the clean accuracy in that model is being traded for a sort of overfitting to the noise-type distribution. Similarly, if the method presented also lowers clean accuracy, what's to guarantee that the improved corruption accuracy isn't simply being traded off by the clean accuracy loss?
> >
> > Another way to think about this is that clean accuracy needs to be controlled for: for instance, checking whether models trained with the method are more "effectively robust" [1], which verifies the OOD robustness of a model for a given clean accuracy (with the expectation that higher-accuracy models will have higher OOD absolute accuracy).
> >
> > [1] https://openreview.net/forum?id=HyxPIyrFvH
> >
> > Minor comments:
> > > We have shown this is indeed true in Fig 7. in Appendix C.2. (left).
> > Thanks for pointing this out. In this fig (now numbered 8 I believe), why does the standard baseline seem to mostly use high frequencies, whereas the high-pass filtered seem to use low and high frequencies? (is the legend wrong?)
> >
> > I believe the changes to the paper made by the authors have improved its presentation. While I still don't recommend acceptance (due to the main methodological concern above), I would like to recognize their hard work with an updated score of 3

---

> > > ### Author Response · Authors · 2021-11-30
> > > **Response**
> > >
> > > Dear reviewer,
> > >
> > > Thank you for your response and questions.
> > >
> > > > Similarly, if the method presented also lowers clean accuracy, what's to guarantee that the improved corruption accuracy isn't simply being traded off by the clean accuracy loss?
> > >
> > > We would like to note here that SFS regularization is a principled method that is not immediately equivalent to any noise augmentation (e.g. unlike Gaussian noise regularization). We believe that this enables SFS regularization to help in diverse distortion scenarios that target the Fourier-sensitivity of models whereas noise augmentation methods may not be expected to. It may also be easier to characterize scenarios of data-shift where SFS regularization can help due to its principled approach, whereas it is not straightforward to characterize where methods like AugMix are expected to work well.
> > >
> > > >why does the standard baseline seem to mostly use high frequencies, whereas the high-pass filtered seem to use low and high frequencies? (is the legend wrong?)
> > >
> > > We would like to clarify that high or low *sensitivity* to a frequency is not necessarily equivalent to *use* of that frequency, although they are likely correlated. Further, we hypothesize that standard training on CIFAR10 leads to a high-frequency sensitivity as high-frequencies maybe needed to maximize clean accuracy on this dataset. This may be why standard-training/ERM leads to a high-frequency sensitivity on CIFAR10/100 but not on SVHN.
> > >
> > > We will try to do additional experiments to better understand the clean/robust acc. behavior of methods and respond if the forum allows further comments from authors.

---

### Official Review · Reviewer_pK3z · 2021-11-07

**Correctness:** 4
**Technical Novelty And Significance:** 4
**Empirical Novelty And Significance:** 3
**Recommendation:** 6
**Confidence:** 3

**Main Review:**

Paper strengths:
Compared with training on fourier-filtered data or applying complex data augmentations, the proposed method is a simpler (and principled) approach to prevent neural networks to bias towards superficial features of the training data.  The motivation and the method sections are well written.  These experiments indicate its effectiveness in learning global structures of the data.  The results in the robustness against the fourier filtering are convincing.

My concerns:
My major concern is about the significance of the experimental results for data corruptions.  This looks to be the main experiment section and AugMix [1] outperforms standard training in all cases but the proposed method does not.  I would agree that the proposed method could be simpler and extendable to other domains but additional experiments were not provided. (only cifar10 & cifar100)
In addition, I’m not sure about the significance of outperforming “medium/high pass filtered” methods when there are a few data corruption cases the standard training (lambda=0) outperforms the regularized training. (6/24 cases with LSF in CIFAR100, for example).  While the regularization term is present, the original cross entropy loss still operates on raw images so the model might still be picking up high frequency local features under LSF regularization.  The sensitivity to this weighting hyperparameter between the two loss terms is not discussed.    If I’m not mistaken, baseline methods that are trained on filtered data cannot take advantage of those local features.

A minor note:
I suspect AugMix [1] won’t perform as well as SFS in the patch-shuffle experiment but it was not provided.  Is AugMix not applicable in this setting?

[1] Dan Hendrycks, Norman Mu, Ekin D. Cubuk, Barret Zoph, Justin Gilmer, and Balaji Lakshminarayanan. AugMix: A Simple Data Processing Method to Improve Robustness and Uncertainty. arXiv:1912.02781 [cs, stat], February 2020. URL http://arxiv.org/abs/1912. 02781. arXiv: 1912.02781.

**Summary Of The Paper:**

This paper proposes a novel spatial frequency regularization technique that improves the robustness of training neural networks against superficial fourier statistics in a dataset.  In the loss function, It adds a regularization term that is based on the Fourier-transformed input-Jacobian.  This term could be customized such that the trained model could ignore (be insensitive to) features with specific frequency in the dataset.   The authors define spatial frequency sensitivity using input-Jacobian in Fourier space.
Although this paper focuses on datasets with images, the method is extendable to any n-dimensional data.  Empirically, the method is evaluated for its robustness against fourier filtering, corruptions, and image patches shuffling in CIFAR10 & CIFAR100 datasets.  It shows better results than other baselines in maintaining the classification accuracy of a model against fourier filtering and image patches shuffling.  There do seem to be slight performance gaps from the SOTA AugMix method in image corruptions.  However, the method is considered to be simpler than AugMix and still outperforms other baselines in many cases.   Experiments demonstrate that the proposed method could learn global features present in a dataset.

**Summary Of The Review:**

As the extendibility to any n-dimensional data is its big advantage over those complex data augmentation methods, I am interested in seeing some results in other domains.  Given how well AugMix performs against the standard training procedure in data corruptions, I may increase my score if additional experiments are provided to reflect its ease of use/extendibility over methods such as AugMix.

---

> ### Author Response · Authors · 2021-11-23
> **Response**
>
> Dear reviewer,
>
> Thanks for your valuable feedback and comments.
>
> > As the extendibility to any n-dimensional data is its big advantage over those complex data augmentation methods, I am interested in seeing some results in other domains
>
> While our method is extendible to any n-dimensional data we restrict ourselves to visual (2D) input in this paper as the emphasis is on spatial frequency. Exploring how our method performs in other tasks and domains is an interesting direction for future work.
>
> > While the regularization term is present, the original cross entropy loss still operates on raw images so the model might still be picking up high frequency local features under LSF regularization
>
> We agree that decreasing the Fourier-sensitivity of the model to certain frequencies does not make those frequencies completely unavailable to the model in the way filtering would. We discuss this point in Appendix D.3 (previously Section 4.6).
>
> > I suspect AugMix [1] won’t perform as well as SFS in the patch-shuffle experiment but it was not provided. Is AugMix not applicable in this setting?
>
> Thanks for the suggestion, we have added AugMix to this benchmark (Table 5). Indeed, the results suggest AugMix does not use global features as much as the SFS-regularized and adversarially trained models.
>
> > The sensitivity to this weighting hyperparameter between the two loss terms is not discussed
>
> We observed a simple trade-off between the SFS-regularization hyperparameter 	and clean accuracy. The more strictly (higher hyperparameter) we regularized the 	model to have the desired SFS, the more the clean accuracy dropped. We found 	that it was not necessary to use high values of this parameter to nudge 		training towards a solution that had the desired frequency characteristics. To keep 	things 	simple, we used a value of 1 for all experiments and datasets, lower did not 	always achieve the desired SFS. We have added this point to the paper in the 	Experiments section.

---

> > ### Comment · Reviewer_pK3z · 2021-11-30
> > **Response to Authors**
> >
> > Thank you for responding to my concerns and updating the presentation.
> > I've read the response as well as other reviews.   The additional experiments in SVHN and patch-shuffled tests do strengthen the presentation so I've updated my score.  However, the drop in clean accuracy observed by reviewer GEyu does indicate the sensitivity of this regularization method.  This is in accordance with my concern about the hyperparameter sensitivity.  This is still a concern as other augmentation methods such as AugMix still preserve the clean accuracy.

---

> > > ### Author Response · Authors · 2021-11-30
> > > **Response**
> > >
> > > Dear reviewer,
> > >
> > > Thank you for your response. FYI, we have responded to comments from Reviewer GEyu.

---

### Author Response · Authors · 2021-11-23
**Major updates**

Dear reviewers,

Thank you for your thoughtful feedback and comments on our submission. We are glad that you appreciated the novelty of our principled Fourier-sensitivity measure and regularizer, the “interesting insight” provided by its derivation applying the chain rule to the input-Jacobian, and general applicability to any differentiable computer vision model. Our paper adds to existing evidence that CNNs develop sensitivity to Fourier modes and understanding this may aid efforts to improve generalization.

A common concern among reviewers was the performance of our proposed regularizer in relation to AugMix in the CIFAR10-C/CIFAR100-C benchmark. We note that AugMix is a data augmentation method specially designed for corruptions in CIFAR10-C/CIFAR100-C, whereas our proposed Fourier-sensitivity measure and regularizer are general techniques for studying computer vision models that can also help improve robustness to these corruptions. Further, we show that AugMix’s high performance does not translate to high performance on a new class of corruptions (Section 4.3) as well as to corruptions in CIFAR10-C when applied to SVHN (Section 4.4).

We understand our previous presentation of the overall approach and results may not have made this easy to appreciate, as noted by Reviewer womq. Hence, considering all reviewers’ suggestions and feedback, we have made the following updates to our submission:

Updated experiments:

1. **We added a new benchmark on a recently proposed class of data-agnostic corruptions (Section 4.3 in revised submission) to demonstrate the ability of spatial frequency regularization to be more robust than AugMix. The generality of our approach allows it to be useful in such new settings. AugMix training does not provide the same level of robustness to these corruptions.  We evaluated on SVHN, CIFAR10 and CIFAR100.**

2. We added results for SVHN-C in Section 4.4, which we generated by applying corruptions in CIFAR10-C to the SVHN dataset, to evaluate these corruptions on a new dataset. We observed that while SFS regularization can improve robustness due to its general approach, AugMix does not have the same level of high performance it has on CIFAR10-C. In fact, SFS regularization outperforms AugMix on many corruptions and has competitive performance on the rest.

3. We added evaluations on CIFAR100 to the Fourier-filtering and Patch-shuffle benchmarks to increase the diversity of results. We found SFS regularization again improved robustness while learning global features.

4. We added SFS plots of models trained on SVHN (Appendix C.3). In contrast to CIFAR10/CIFAR100 and Imagenet, standard trained models on SVHN have a low-frequency sensitivity, a new observation that suggests a strong dataset dependence of spatial frequency sensitivity. This challenges the common belief that standard training leads always to a high-frequency bias.

5. We added AugMix to the Patch Shuffle benchmark (Table 5) at the suggestion of Reviewer pK3z. The results indeed suggest that AugMix does not use global features as much as the SFS-regularized and adversarially trained models.

6. Minor update: We updated the gaussian-noise augmented model in all benchmarks. Previously, we added gaussian-noise before input-normalization of training images. We updated it to add gaussian-noise after input-normalization so the mean and std. deviation of the additive gaussian noise were preserved. While this slightly improved performance as well as decreased the model’s high-frequency sensitivity (Fig 3), its performance relative to other methods and overall interpretation of results did not change.

Updates to presentation:

1. We updated the paper title to “Fourier Sensitivity and Regularization of Computer Vision” to reflect the generality of the proposed spatial frequency sensitivity method.

2. We updated the abstract and introduction to better reflect the need to study the structural sensitivity of computer vision models to Fourier-basis components as well as the generality of the proposed method beyond just robustness to CIFAR10-C/CIFAR100-C (in accordance with author guidelines (FAQ): https://iclr.cc/Conferences/2021/AuthorGuide).

3. To make space for new benchmarks, we moved the CIFAR100-C benchmark (previously Table 3.), which was similar in results to CIFAR10-C, to Appendix F.2.

4. We removed results for models trained on medium-pass and high-pass filtered data in CIFAR10/100-C benchmarks as their results were very low and not meaningful. We retain the discussion about Fourier-filtered training (previously Section 4.6) but moved it to Appendix D.3 to create space for new benchmarks.

5. Fixed typos and rephrased some sentences for conciseness.

We have highlighted all major updates of the manuscript in blue. We hope these updates make it easier to see the benefits of the proposed methods. Thank you for your consideration.

---

### Decision · Program_Chairs · 2022-01-20

**Decision:**

Reject

**Comment:**

The reviewers all generally appreciated the idea in the paper. However, the nature of this contribution necessitates an empirical evaluation, and the reviewers generally found this to not be sufficient convincing. My assessment is that this idea can likely result in a successful publication, but will require additional empirical evaluation and analysis as suggested by reviewers. While the authors did add some additional results during the response period, they do not seem to be sufficient to fully address reviewer concerns.